# Development and validation of an early risk-stratification model for hemophagocytic lymphohistiocytosis in severe fever with thrombocytopenia syndrome

Yiqiang Hu[1☉], Qionghan He[2☉], Yuxiao Liu[3], Jiaqi Kong[1], Jianting He[3], Rui Liu[3], Wenting Li[2]*, Chuanlong Zhu[1,3]*

1 Department of Infectious Disease, The First Affiliated Hospital with Nanjing Medical University, Nanjing, China, 2 Department of Infectious Disease, The First Affiliated Hospital with Anhui Medical University, Hefei, China, 3 Department of Infectious and Tropical Diseases, The Second Affiliated Hospital of Hainan Medical University, Haikou, China

☉ These authors contributed equally to this work.
* Wtl9911002@163.com (WL); zhuchuanlong@jsph.org.cn (CZ)

## Abstract

### Objective

To develop and validate a model for early risk stratification of secondary hemophagocytic lymphohistiocytosis (HLH) in patients with severe fever with thrombocytopenia syndrome (SFTS).

### Methods

This retrospective cohort included adults with laboratory-confirmed SFTS admitted to The First Affiliated Hospital with Nanjing Medical University between January 2019 and July 2024. Predictor variables were derived from clinical and laboratory data obtained within 3 days after virologic confirmation, corresponding to the predefined early-evaluation window of the study. HLH status (binary outcome) was defined using the entire-course HScore (≥170), calculated from the worst available values over the clinical course; HLH-2004 criteria and early-window HScore distributions were summarized descriptively to provide transparent outcome accounting. Twenty-eight candidate predictors entered LASSO with Boruta refinement, and retained variables were used to construct a multivariable logistic model and nomogram. Model performance was evaluated by discrimination, calibration, and decision-curve analysis in the derivation cohort, an internal validation set, and an external cohort of 60 patients from The First Affiliated Hospital with Anhui Medical University.

### Results

Among 249 patients (152 non-HLH, 97 HLH), HLH was associated with higher peak temperature, longer fever, more lymphadenopathy/splenomegaly and neurological

**Data availability statement:** De-identified participant data supporting the findings of this study are not publicly available due to participant privacy and institutional restrictions. Data are available upon reasonable request from the Ethics Committee of The First Affiliated Hospital with Nanjing Medical University, Nanjing, China (email: jsphkj@163.com). Requests will be reviewed for compliance with institutional ethics requirements and applicable regulations, and may require a data use agreement.

**Funding:** This work was supported by the Science and Technology Plan of Hainan Province (Clinical Research Center; grant LCYX202306 to C.Z. and grant LCYX202408 to R.L.), the Natural Science Foundation of Jiangsu Province (grant BK20241982 to C.Z.), and grants from the Hainan Province Clinical Medical Center to C.Z. The funders had no role in study design, data collection and analysis, decision to publish, or preparation of the manuscript.

**Competing interests:** The authors have declared that no competing interests exist.

symptoms, and more severe thrombocytopenia, hypertriglyceridemia, hypofibrinogenemia, hyperferritinemia, higher viral load, elevated muscle/liver enzymes and LDH, and coagulopathy (all $p < 0.05$). LASSO–Boruta identified six routinely available predictors—peak temperature, splenomegaly, fever duration, triglycerides, fibrinogen, and ferritin. The model showed LR $\chi^2 = 214.82$ ($p < 0.0001$), $R^2 = 0.784$, C-index = 0.962, Dxy = 0.923, with near-perfect calibration in derivation. In internal validation, discrimination remained near-perfect (AUC 0.997, 95% CI 0.989–1.000); mild miscalibration was corrected by intercept-and-slope recalibration, and decision curves showed net benefit across wide thresholds. External validation (n = 60) confirmed excellent discrimination (AUC 0.907, 95% CI 0.835–0.980), slight miscalibration resolved by recalibration, and preserved net benefit across most thresholds.

## Conclusions

A simple model based on early clinical and laboratory variables supports risk stratification for HLH in SFTS and may facilitate closer monitoring, repeated HLH assessment, and timely individualized management.

### Author summary

Severe fever with thrombocytopenia syndrome (SFTS) is a tick-borne viral infection associated with substantial mortality, and secondary hemophagocytic lymphohistiocytosis (HLH) is one of its most severe complications. Early risk recognition and timely diagnostic reassessment for HLH in patients with SFTS remain significant clinical challenges. In this study, we developed a model based on routinely available clinical and laboratory variables collected during an early evaluation window. The model showed high accuracy and potential clinical utility for early risk stratification. Rather than replacing established diagnostic frameworks, it may help clinicians identify high-risk patients who warrant closer monitoring, repeat HLH assessment, and earlier escalation of care.

## Introduction

Severe fever with thrombocytopenia syndrome virus (SFTSV), a novel tick-borne bunyavirus, is the etiologic agent of severe fever with thrombocytopenia syndrome (SFTS). Human infection with SFTSV and its pathogenicity were first identified and confirmed in China [1]. SFTS is predominantly a tick-borne zoonosis with geographically focal endemicity in China and other East Asian regions. Clinically, SFTS typically presents with acute fever, thrombocytopenia, and leukopenia, often accompanied by multiorgan dysfunction; a subset of patients develops coagulopathy, including disseminated intravascular coagulation [2,3]. HLH is a syndrome of pathologic immune overactivation characterized by prolonged high-grade fever, cytopenias,

hyperferritinemia, and coagulopathy. The syndrome progresses rapidly and, if untreated, is associated with high mortality [4,5]. HLH encompasses genetically determined primary (familial) disease and secondary forms triggered by infection, malignancy, or autoimmune/rheumatologic disorders [4,6]. SFTSV infection can precipitate sHLH, and existing data indicate that HLH is a life-threatening complication of SFTS associated with poor clinical outcomes and increased mortality [7]. However, prospective data to support risk stratification for HLH in the setting of SFTSV infection are scarce, and available early prediction tools and objective biomarkers remain inadequately developed and validated [5,8]. HScore is a composite clinical scoring system developed to estimate the probability of reactive hemophagocytic syndrome on the basis of features such as fever, organomegaly, cytopenias, ferritin, triglycerides, and fibrinogen; however, it was not specifically developed for SFTS and may rely on data that accumulate over time. Accordingly, early recognition of patients at high risk for SFTS-associated HLH using variables obtainable soon after virologic confirmation could facilitate closer monitoring, earlier diagnostic reassessment, and more individualized management. In this study, we retrospectively evaluated clinical characteristics and laboratory parameters obtained within 3 days after virologic confirmation in hospitalized patients with SFTS, identified factors associated with HLH, and developed a multivariable model to assess its utility for early risk stratification of SFTS-associated HLH [8].

## Materials and methods

### Ethics statement

The study protocol was reviewed and approved by the Ethics Committee of The First Affiliated Hospital with Nanjing Medical University (approval No. 2025-SR-955) and was conducted in accordance with the Declaration of Helsinki. This study involved secondary analysis of de-identified clinical data collected during routine care. The requirement for written informed consent was waived by the institutional review boards in view of the retrospective study design and minimal risk to participants.

### Design and setting

This single-center retrospective study included 249 patients with laboratory-confirmed SFTS who were hospitalized in the Department of Infectious Diseases at The First Affiliated Hospital with Nanjing Medical University between January 2019 and July 2024, as identified via the hospital electronic medical record system. For external validation, we retrospectively identified an independent cohort of 60 SFTS patients admitted to The First Affiliated Hospital with Anhui Medical University between January 2020 and January 2024 using the same eligibility criteria.

### Participants

Inclusion criteria were as follows: (i) laboratory confirmation of SFTSV infection by nucleic acid testing [1,9]; (ii) age ≥ 18 years. Exclusion criteria were: (i) a preexisting hematologic disorder; (ii) active malignancy or rheumatic immune-mediated disease [5]; (iii) missing core clinical or laboratory data required for the predefined early-evaluation window (within 3 days after virologic confirmation); (iv) discharge at the patient's request within 72 hours. For model development and validation, HLH status (outcome) was defined using the entire-course HScore (≥170), calculated using the worst available values over the clinical course. HLH-2004 criteria were collected when available and summarized descriptively (Supporting Information S2 and S3 Tables) to provide a transparent accounting of outcome adjudication; these criteria were not used as a formal requirement for binary outcome classification. Accordingly, patients were classified into a Non-HLH cohort (n = 152) or an HLH cohort (n = 97) [4,8]. Early-window HScore distributions were also summarized descriptively to document how frequently patients in the final HLH group already met HScore criteria during the predefined early-evaluation period. Temporal anchoring for both predictor ascertainment and early HScore accounting was based on the first 3 days after virologic confirmation.

## Variables and definitions

Clinical data for both cohorts were retrieved from the hospital electronic medical record system, including interval from symptom onset to admission, peak preadmission body temperature, duration of fever, lymphadenopathy, splenomegaly, and neurologic manifestations (e.g., altered mental status, headache, vomiting). The interval from symptom onset to admission was recorded and was used as an operational summary of symptom duration at the time early data were captured for model development. In addition, laboratory parameters obtained within 3 days of virologic confirmation were recorded, including hemoglobin, leukocyte and neutrophil counts, platelet count, triglycerides, creatine kinase, alanine aminotransferase, aspartate aminotransferase, lactate dehydrogenase, coagulation indices, and SFTSV blood viral load. This same predefined early-evaluation window was used for descriptive early HScore accounting. The exact laboratory report-return time was not consistently extractable from the retrospective record; however, in routine clinical practice, positive nucleic acid results were typically returned within approximately half a day of virologic confirmation. Accordingly, the recorded time of virologic confirmation was used as the operational temporal anchor for early predictor ascertainment and early HScore assessment. Quantitative values exceeding the upper limit of detection were truncated to that limit. For the small proportion of missing continuous data, imputation followed established methods: approximately normally distributed variables were imputed using the mean, whereas non-normally distributed variables were imputed using the median. Continuous variables with large order-of-magnitude variation were converted into categorical variables according to published evidence and international practice [10–16]. For laboratory variables summarized in Table 1, the most abnormal value within this 3-day window was used (maximum or minimum as appropriate).

## Statistical analysis

Statistical analyses were conducted in R (version 4.5.1). Baseline characteristics were tabulated. Continuous variables were expressed as mean ± standard deviation or median (interquartile range) and compared between groups using the independent-samples t-test or Mann–Whitney U-test, as appropriate. Categorical variables were summarized as counts (percentages) and compared using the $\chi^2$ test or Fisher's exact test when expected cell counts were <5. Two-sided P values were tabulated to describe differences in baseline characteristics between groups. Variable selection was initially performed using least absolute shrinkage and selection operator (LASSO) regression. To assess the robustness of selection and mitigate method-specific bias, we additionally applied the Boruta algorithm. Predictors retained by both approaches were defined as the final variable set for inclusion in the multivariable logistic regression model.

The full cohort was randomly partitioned (85:15) into a training set and an internal validation set, and the multivariable logistic regression model was fitted in the training set using the final selected predictors. Model performance was evaluated in both the internal validation set and the external validation cohort using the derived multivariable combined prediction model. Discrimination was quantified by the area under the receiver operating characteristic curve (AUC) and the Brier score. Calibration was assessed using calibration plots. Clinical utility was evaluated with decision curve analysis (DCA). A two-sided P value <0.05 was considered statistically significant.

## Results

A total of 249 hospitalized patients with confirmed SFTS were included in the analysis (Table 1). The median age was 63 years (IQR, 55–69), and 47% were male. The median interval from symptom onset to admission was 7 days (IQR, 5–8), which served as an operational summary of symptom duration at the time data were captured for model development. Most patients had persistent fever (median duration, 7 days; IQR, 6–9). Lymphadenopathy and splenomegaly were present in 53% and 6% of patients, respectively. Neurologic manifestations—most commonly headache, psychomotor slowing, and altered mental status—were recorded. Gastrointestinal involvement was frequent, with vomiting or diarrhea reported in 49% and concurrent vomiting and diarrhea in 13%. Hematologically, thrombocytopenia was prominent, with a median platelet count of $50 \times 10^9$/L (IQR, 34–68). Hypertriglyceridemia, marked hyperferritinemia, and coagulation abnormalities

**Table 1. The demographic and clinical characteristics of patients with SFTS with and without HLH.**

| Parameters | Total (N = 249) | Non-HLH (n = 152) | HLH (n = 97) | p |
|---|---|---|---|---|
| Age, y, median (IQR) | 63 (55, 69) | 63 (52, 69) | 63 (58, 71) | 0.199 |
| Sex, male, n (%) | 118 (47%) | 76 (50%) | 42 (43%) | 0.367 |
| From onset to admission, days, median (IQR) | 7 (5, 8) | 7 (5, 7) | 7 (6, 8) | 0.02 |
| Underlying disease, n (%) | | | | 0.618 |
| One | 58 (23%) | 33 (22%) | 25 (26%) | |
| Two | 16 (6%) | 8 (5%) | 8 (8%) | |
| Three or more | 4 (2%) | 3 (2%) | 1 (1%) | |
| Maximal temperature, °C, n (%) | | | | < 0.001 |
| ≤ 38 °C | 27 (11%) | 24 (16%) | 3 (3%) | |
| 38.1–39.0 °C | 135 (54%) | 86 (57%) | 49 (51%) | |
| ≥ 39.1 °C | 87 (35%) | 42 (28%) | 45 (46%) | |
| Fever, days, median (IQR) | 7 (6, 9) | 7 (6, 9) | 9 (7, 10) | < 0.001 |
| Myalgia, n (%) | 84 (34%) | 49 (32%) | 35 (36%) | 0.625 |
| Lymphadenopathy, n (%) | 132 (53%) | 71 (47%) | 61 (63%) | 0.018 |
| Splenomegaly, n (%) | 14 (6%) | 3 (2%) | 11 (11%) | 0.004 |
| Neurological symptoms, n (%) | | | | 0.002 |
| Headache | 66 (27%) | 36 (24%) | 30 (31%) | |
| Psychomotor slowing | 47 (19%) | 20 (13%) | 27 (28%) | |
| Altered mental status | 6 (2%) | 3 (2%) | 3 (3%) | |
| Gastrointestinal symptoms, n (%) | | | | 0.1 |
| Vomiting or diarrhea | 123 (49%) | 75 (49%) | 48 (49%) | |
| Vomiting and diarrhea | 33 (13%) | 15 (10%) | 18 (19%) | |
| Platelet count, $10^9$/L, median (IQR) | 50 (34, 68) | 56.5 (40.75, 75) | 40 (29, 58) | < 0.001 |
| Hemoglobin, g/L, mean ± SD | 124.96 ± 17.32 | 126.12 ± 16.42 | 123.14 ± 18.57 | 0.199 |
| Leukocyte count, $10^9$/L, median (IQR) | 2.19 (1.62, 3.36) | 2.22 (1.58, 3.14) | 2.12 (1.63, 3.47) | 0.868 |
| Neutrophil count, $10^9$/L, median (IQR) | 1.16 (0.78, 1.87) | 1.08 (0.78, 1.73) | 1.32 (0.78, 2.22) | 0.125 |
| Triglyceride, mmol/L, median (IQR) | 2.29 (1.69, 3.13) | 2.03 (1.50, 2.67) | 2.92 (2.08, 3.78) | < 0.001 |
| Fibrinogen, g/L, mean ± SD | 2.28 ± 0.5 | 2.38 ± 0.51 | 2.12 ± 0.44 | < 0.001 |
| Ferritin, ng/ml, n (%) | | | | < 0.001 |
| < 2000 ng/mL | 134 (54%) | 128 (84%) | 6 (6%) | |
| 2000–6000 ng/mL | 61 (24%) | 14 (9%) | 47 (48%) | |
| > 6000 ng/mL | 54 (22%) | 10 (7%) | 44 (45%) | |
| Viral load, n (%) | | | | < 0.001 |
| < $10^4$ | 26 (10%) | 22 (14%) | 4 (4%) | |
| $10^4$–$10^6$ | 127 (51%) | 86 (57%) | 41 (42%) | |
| > $10^6$ | 96 (39%) | 44 (29%) | 52 (54%) | |
| Creatine Kinase[a], U/L, n (%) | | | | < 0.001 |
| Normal | 102 (41%) | 82 (54%) | 20 (21%) | |
| Mildly elevated | 74 (30%) | 40 (26%) | 34 (35%) | |
| Moderately elevated | 54 (22%) | 22 (14%) | 32 (33%) | |
| Severely elevated | 19 (8%) | 8 (5%) | 11 (11%) | |
| AST[b], U/L, n (%) | | | | < 0.001 |
| Normal | 6 (2%) | 6 (4%) | 0 (0%) | |
| Mildly elevated | 92 (37%) | 72 (47%) | 20 (21%) | |
| Moderately elevated | 116 (47%) | 62 (41%) | 54 (56%) | |
| Severely elevated | 35 (14%) | 12 (8%) | 23 (24%) | |

*(Continued)*

**Table 1.** (Continued)

| Parameters | Total (N = 249) | Non-HLH (n = 152) | HLH (n = 97) | p |
|---|---|---|---|---|
| ALT[c], U/L, n (%) | | | | 0.013 |
| Normal | 51 (20%) | 39 (26%) | 12 (12%) | |
| Mildly elevated | 143 (57%) | 87 (57%) | 56 (58%) | |
| Moderately elevated | 52 (21%) | 25 (16%) | 27 (28%) | |
| Severely elevated | 3 (1%) | 1 (1%) | 2 (2%) | |
| AST/ALT, median (IQR) | 2.3 (1.78, 2.91) | 2.07 (1.63, 2.57) | 2.65 (2.18, 3.54) | < 0.001 |
| LDH[d], U/L, n (%) | | | | < 0.001 |
| Normal | 9 (4%) | 9 (6%) | 0 (0%) | |
| Mildly elevated | 67 (27%) | 54 (36%) | 13 (13%) | |
| Moderately elevated | 123 (49%) | 66 (43%) | 57 (59%) | |
| Severely elevated | 50 (20%) | 23 (15%) | 27 (28%) | |
| PT[e], s, n (%) | | | | 0.39 |
| Normal | 232 (93%) | 140 (92%) | 92 (95%) | |
| Mildly elevated | 14 (6%) | 10 (7%) | 4 (4%) | |
| Moderately elevated | 1 (0%) | 0 (0%) | 1 (1%) | |
| Severely elevated | 2 (1%) | 2 (1%) | 0 (0%) | |
| APTT[f], s, n (%) | | | | < 0.001 |
| Normal | 90 (36%) | 72 (47%) | 18 (19%) | |
| Mildly elevated | 108 (43%) | 59 (39%) | 49 (51%) | |
| Moderately elevated | 38 (15%) | 15 (10%) | 23 (24%) | |
| Severely elevated | 13 (5%) | 6 (4%) | 7 (7%) | |
| TT[g], s, n (%) | | | | < 0.001 |
| Normal | 36 (14%) | 28 (18%) | 8 (8%) | |
| Mildly elevated | 114 (46%) | 86 (57%) | 28 (29%) | |
| Moderately elevated | 38 (15%) | 16 (11%) | 22 (23%) | |
| Severely elevated | 61 (24%) | 22 (14%) | 39 (40%) | |

Using clinical features and the most abnormal laboratory values within 3 days after virologic confirmation. Laboratory parameters summarize the most abnormal values within 3 days after virologic confirmation: ferritin, triglycerides, AST, ALT, creatine kinase, lactate dehydrogenase, viral load, thrombin time, activated partial thromboplastin time, and prothrombin time represent maxima; platelet count, fibrinogen, neutrophil count, hemoglobin, and white blood cell count represent minima.

Severity was categorized as Normal/ Mild elevation/ Moderate elevation/ Severe elevation using the following thresholds:

[a] CK (U/L; sex-specific): males 50–310; > 310–930; > 930–3,100; > 3,100; females ≤200; > 200–600; > 600–2,000; > 2,000. [bc] ALT or AST (U/L): ≤ 40; > 40–120; > 120–399.9; ≥ 400. [d] LDH (U/L): ≤ 245; > 245–489.9; 490–1,224.9; ≥ 1,225. [e] Prothrombin time (PT, s): ≤ 13; > 13–15.9; 16–18.9; ≥ 19. [f] Activated partial thromboplastin time (APTT, s): ≤ 35; > 35–44.9; 45–54.9; ≥ 55. [g] Thrombin time (TT, s): ≤ 18; > 18–21.9; 22–25.9; ≥ 26.

were frequently observed. Most patients had detectable SFTSV viremia $\geq 10^4$ copies, and elevations in creatine kinase, aminotransferases, and lactate dehydrogenase were common. Outcome accounting based on HScore and HLH-2004 criteria, including distributions within 3 days after virologic confirmation and over the entire clinical course, is provided in Supporting Information S1–S3 Tables. Notably, at the early evaluation point, 83 of 97 patients (85.6%) in the final HLH group already met the HScore threshold for HLH (S1 Table), indicating that the model should be interpreted as a tool for early risk stratification within a clinically relevant window rather than as proof of preclinical prediction. In the overall cohort, 83 of 249 patients (33.3%) met the HScore threshold within 3 days after virologic confirmation, whereas 97 of 249 (39.0%) met the threshold over the entire clinical course.

The Non-HLH cohort comprised 152 patients. The median interval from symptom onset to admission was 7 days (IQR, 5–7). Marked hyperthermia was uncommon; 73% had a maximum temperature ≤39.0°C. The median duration of

fever was 7 days (IQR, 6–9). Lymphadenopathy was present in 47%, whereas splenomegaly was rare (2%). Neurologic involvement was infrequent, with psychomotor slowing in 13% and altered mental status in 2%. The median platelet count was 56.5 × 10⁹/L (IQR, 40.75–75). Triglyceride levels were lower, with a median of 2.03 mmol/L (IQR, 1.50–2.67), whereas fibrinogen was relatively preserved (mean 2.38 g/L). Serum ferritin was < 2000 ng/mL in most patients (84%), and only 7% had levels >6000 ng/mL. Viral load also tended to be lower, with 14% of patients having <10⁴ copies. Moderate-to-severe elevations in creatine kinase, AST, and LDH were less frequent than in the HLH cohort. Coagulation profiles were comparatively preserved, with predominantly normal PT and fewer marked prolongations of APTT and TT.

Relative to the Non-HLH cohort, the HLH cohort had a modestly longer onset-to-admission interval (P = 0.02), higher peak temperature and more prolonged fever duration (both P < 0.001), more frequent lymphadenopathy and splenomegaly (P = 0.018 and P = 0.004, respectively), and more frequent neurologic manifestations (P = 0.002). In the HLH cohort, laboratory abnormalities included lower platelet counts (P < 0.001); higher triglycerides and ferritin with a rightward shift toward ≥2,000 and >6,000 ng/mL (P < 0.001); lower fibrinogen (P < 0.001); higher SFTSV viral load (P < 0.001); greater elevations in CK, AST, ALT, and LDH with a higher AST/ALT ratio (all P ≤ 0.013); and more frequent prolongation of APTT and TT, with no significant difference in PT (P < 0.001 for APTT/TT; P = 0.39 for PT). Overall, the HLH cohort exhibited more profound thrombocytopenia, a heightened inflammatory phenotype, higher levels of viremia, and more severe coagulation derangement. Areas under the receiver operating characteristic curve (AUCs) were calculated to quantify the discriminatory performance of individual early clinical and laboratory variables for HLH complicating SFTS (Fig 1).

LASSO regression was applied to 28 candidate predictors, followed by refinement with the Boruta algorithm, yielding six variables—peak preadmission temperature, splenomegaly, fever duration, triglycerides, fibrinogen, and ferritin—for inclusion in the multivariable logistic regression model. A multivariable logistic regression model incorporating the six selected predictors was developed and presented as a nomogram (Fig 2). Calibration was assessed using calibration curves (Fig 3). In the overall cohort (n = 249), the likelihood ratio chi-square was 214.82 (P < 0.0001) with R² = 0.784. The model demonstrated excellent discrimination, with a C-index of 0.962 and a Dxy of 0.923, indicating robust separation between low- and high-risk patients.

To evaluate the performance of the combined prediction model, the cohort of 249 patients was randomly split in an 85:15 ratio into derivation and internal validation sets. Discrimination in the derivation cohort was excellent (Fig 3A), and apparent calibration was near-ideal (Fig 3B), with an intercept close to 0, a calibration slope of 1.00, and an average absolute calibration error (Eavg) of 0.0025. In the internal validation set, discrimination remained nearly perfect (Fig 3C; AUC 0.997, 95% CI 0.989–1.000), but calibration was suboptimal, as reflected by a positive intercept (+4.14) and an inflated slope (2.53), indicating systematic underestimation and inadequate dispersion of predicted risk. Intercept-and-slope recalibration substantially improved concordance between predicted and observed risk (Fig 3D). Decision curve analysis demonstrated a positive net clinical benefit across almost the entire range of reasonable decision thresholds (Fig 3E and 3F).

The external validation cohort was derived from The First Affiliated Hospital with Anhui Medical University, where eligible patients were retrospectively identified using the same criteria and procedures. In this cohort (n = 60), the model (Fig 4A) demonstrated excellent discrimination (AUC 0.907, 95% CI 0.835–0.980). Calibration indicated a small positive intercept (+0.233), consistent with slight underestimation of absolute risk, and a calibration slope of 0.896, suggesting mild overfitting and overdispersion of predicted probabilities. Following intercept-and-slope recalibration, agreement between predicted and observed risk improved substantially (Fig 4B). Consistent with the internal validation, decision curve analysis in the external cohort showed that the model conferred a positive net clinical benefit across nearly the entire range of clinically relevant threshold probabilities (Fig 4C).

## Discussion

Since HLH was recognized as a major complication of SFTS, most published data have consisted of case reports or small retrospective series, and robust evidence on population-level incidence and independent risk factors remains scarce

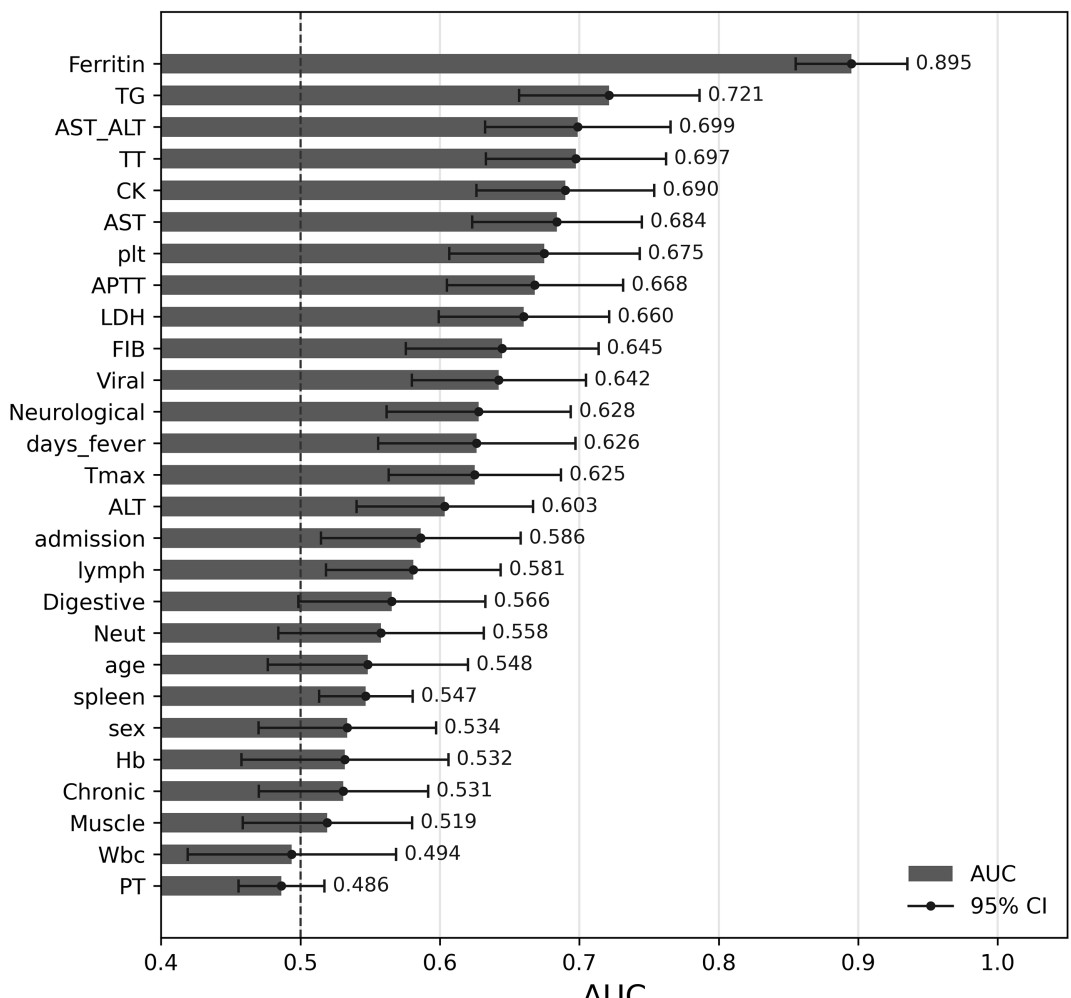

**Fig 1. Comparison of the discriminatory performance of individual early clinical and laboratory variables for HLH complicating SFTS.** Comparison of the areas under the receiver operating characteristic curves (AUCs) for early clinical and laboratory variables measured within 3 days after virologic confirmation. Each bar represents the AUC with its 95% confidence interval for an individual predictor distinguishing SFTS patients who developed HLH from those who did not. Variables are ordered from highest to lowest AUC, with markers at the top indicating the strongest discriminatory ability.

[17–21]. A single-center retrospective study from Korea was the first to systematically evaluate the impact of sHLH on outcomes in SFTS, demonstrating a markedly increased case fatality, with mortality reaching 75% (3 of 4 patients) in a small cohort [7]. In Japan, available data are largely limited to case reports, small case series, and reviews. The overall case-fatality rate of SFTS is approximately 26–30%, and the contribution of concomitant HLH to adverse outcomes has attracted increasing attention [22–25]. In adults, sHLH remains associated with poor outcomes; early case-fatality rates of approximately 30–40% have been reported and are even higher in severe cases and ICU populations [5,26,27]. In China, cohort data on SFTS complicated by HLH have emerged in recent years. One cohort of laboratory-confirmed SFTS cases estimated the incidence of HLH at approximately 7.2% and demonstrated that concomitant HLH was associated with an increased risk of death, with an overall case-fatality rate of about 24.5% in that cohort [19,20]. Taken together, data from

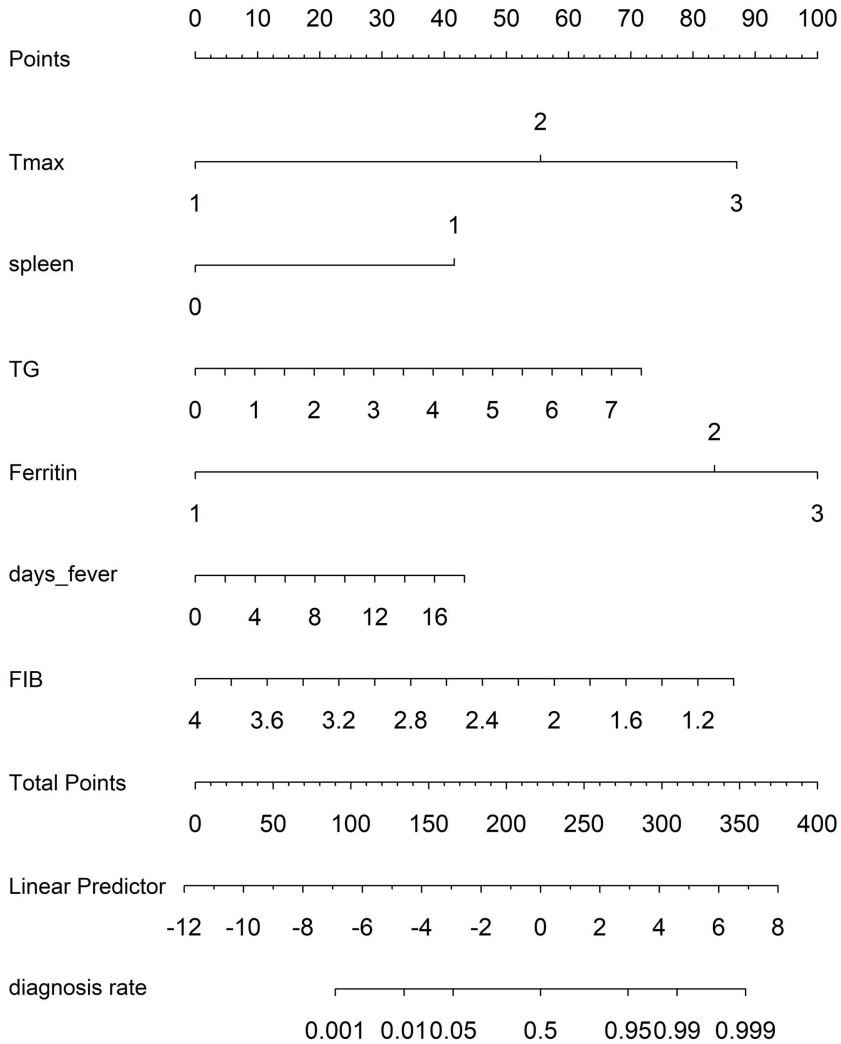

**Fig 2. Nomogram for bedside early risk stratification of secondary hemophagocytic lymphohistiocytosis in patients with severe fever with thrombocytopenia syndrome.** This nomogram is intended for use during the early hospital evaluation of patients with laboratory-confirmed SFTS. After locating the patient's value for each predictor on the corresponding axis, draw a vertical line upward to the Points axis to assign points, sum the points across all predictors, and project the total vertically to the Predicted Probability of HLH scale to estimate individual risk. Higher total points indicate a higher probability of HLH and may justify closer monitoring, repeated HLH assessment, and earlier specialist consultation. Predictor coding: For Tmax, categories 1–3 correspond to ≤38.0 °C, 38.1–39.0 °C, and ≥39.1 °C, respectively. For ferritin, categories 1–3 correspond to <2000 ng/mL, 2000–6000 ng/mL, and >6000 ng/mL, respectively. For splenomegaly, 0 indicates absence and 1 indicates presence.

China, together with parallel observations from Korea and Japan: occurrence of HLH in SFTS is associated with excess mortality and underscores the need for timely recognition, close monitoring, individualized management, and rigorous management of complications to reduce case fatality and improve long-term outcomes. These convergent findings highlight the need to recognize SFTS patients at high risk for HLH at an early stage and to institute targeted immuno-modulatory and intensive supportive care without delay [28–30]. Compared with reports from Korea and Japan, which have largely consisted of small retrospective cohorts, case series, or fatal-case descriptions, and with recent studies from China that have focused primarily on incidence, mortality, or general prognostic factors, the present study differs in three respects. First, it specifically targets HLH risk stratification within a cohort of laboratory-confirmed SFTS. Second, it

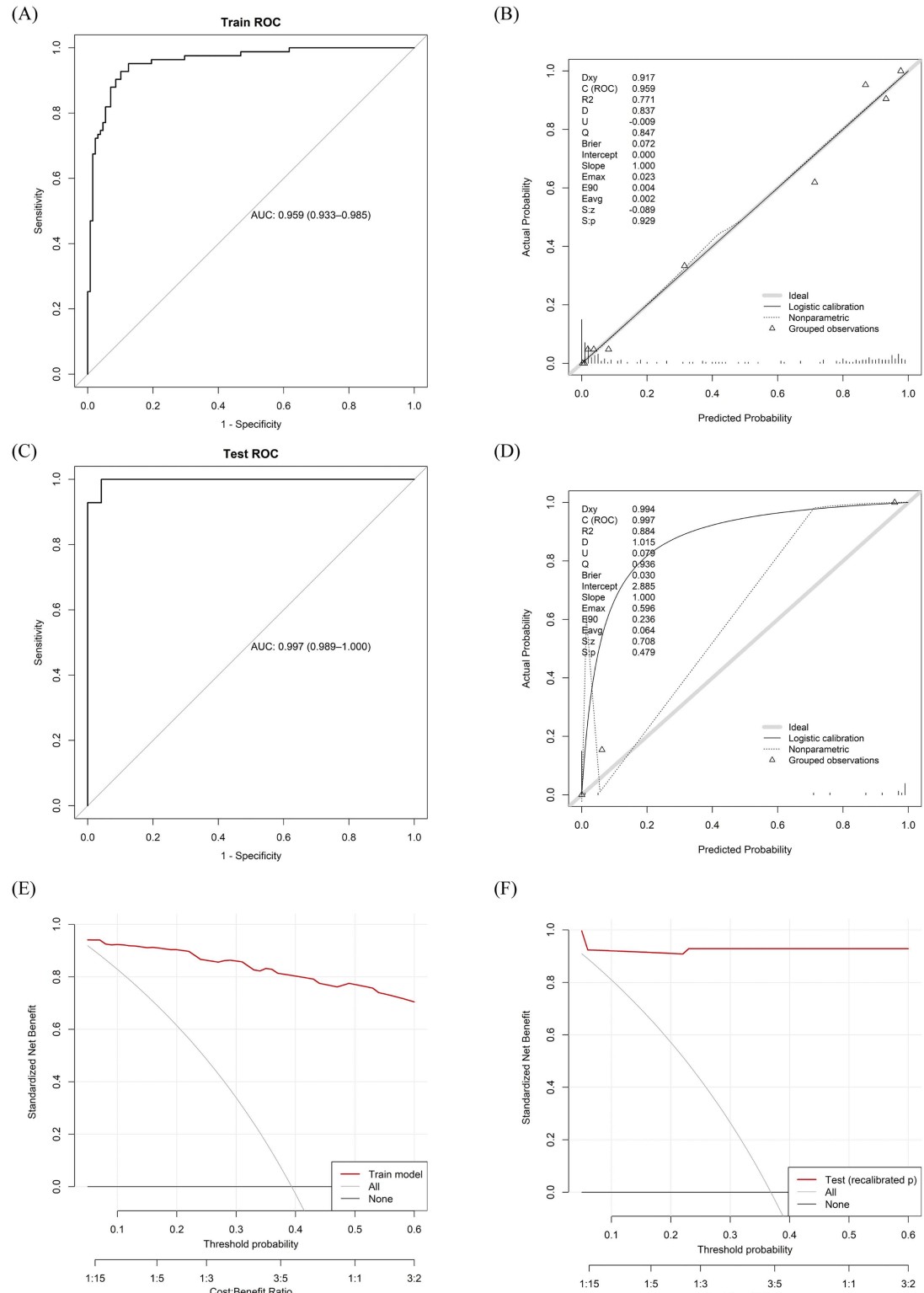

**Fig 3. Discrimination, calibration, and clinical utility of the SFTS–HLH prediction model in the derivation and internal validation cohorts.** Panel **(A)** shows the receiver operating characteristic (ROC) curve of the multivariable model in the training cohort, with the area under the ROC curve (AUC) quantifying discrimination; panel **(B)** displays the corresponding calibration plot comparing predicted and observed risks of secondary HLH in the training

cohort; panel **(C)** shows the ROC curve of the model in the internal validation cohort; and panel **(D)** shows the calibration plot in the internal validation cohort after intercept-and-slope recalibration. Panels **(E)** and **(F)** display decision-curve analyses in the training and internal validation cohorts, illustrating the net clinical benefit of using the prediction model across a range of threshold probabilities compared with "treat-all" and "treat-none" strategies.

uses six routinely accessible variables available during the early clinical course. Third, it includes external validation in an independent cohort. These features may enhance bedside applicability, although broader geographic validation remains necessary. On this basis, we sought to systematically characterize risk factors for HLH complicating SFTS in a real-world regional cohort and to develop a combined prediction model. The goal was to generate a quantitative tool for early risk stratification of SFTS patients at high risk of sHLH, thereby facilitating closer monitoring, repeated HLH assessment, and optimization of individualized management strategies.

The analysis revealed that greater peak preadmission temperature, splenomegaly, extended duration of fever, hypertriglyceridemia, low fibrinogen, and elevated ferritin are risk factors for HLH complicating SFTS. A multivariable model incorporating these variables provided reliable risk estimation for SFTS-associated HLH within the predefined early-evaluation window and demonstrated robust performance in the derivation cohort. The model showed good overall fit ($R^2 = 0.784$), with satisfactory calibration (calibration intercept close to 0 and slope of 1.00) and excellent discrimination (C-index 0.962; Dxy 0.923), indicating near-complete separation between patients at high and low risk of HLH. In both the internal validation set and the external validation cohort, discriminatory performance remained high (AUC 0.997 and 0.907, respectively). Decision curve analysis demonstrated a positive net clinical benefit across nearly the entire range of decision thresholds, supporting the clinical utility of the model and suggesting its generalizability across centers.

Patients with SFTS are at substantial risk for developing sHLH. Once sHLH ensues, the clinical phenotype is characterized by a more intense inflammatory response, more profound thrombocytopenia, more severe coagulopathy, higher circulating viral burden, and worse prognosis. Notably, the six risk indicators incorporated into the model—high peak temperature, splenomegaly, prolonged fever duration, hypertriglyceridemia, hypofibrinogenemia, and marked hyperferritinemia—are readily available early in the hospital course and can be assessed at the bedside with routine clinical and laboratory evaluation. A multivariable model incorporating these parameters may assist early risk stratification during the initial hospital course and could support closer monitoring, repeated HLH assessment, earlier specialist consultation, and timely escalation of supportive management when clinically indicated. Whether use of the model translates into improved survival requires prospective validation.

An important consideration in interpreting the present model is that it was designed to support early risk stratification within a prespecified clinical window rather than to prove that HLH can be identified earlier than with HScore. Notably, 83 of 97 patients (85.6%) in the final HLH group already met HScore criteria at the early evaluation point, indicating that some cases had already evolved to an HLH phenotype by the time early study variables were captured. Accordingly, the incremental advantage of this model over HScore in terms of timeliness cannot be established from the current retrospective dataset. Instead, the principal value of the model may lie in its use of six routinely available variables to facilitate bedside recognition of high-risk SFTS patients, prompt closer reassessment, and support earlier diagnostic escalation. Prospective studies directly comparing this model with HScore in terms of time-to-identification and downstream clinical decision-making are warranted.

From a clinical workflow perspective, the model may be most useful shortly after virologic confirmation and before overt multiorgan failure or unequivocal HLH has been established. Patients flagged as high risk could undergo closer monitoring of ferritin, fibrinogen, triglycerides, cytopenias, coagulation indices, viral load, and organ dysfunction, with earlier hematology consultation and reassessment for evolving HLH. In contrast, once a patient already fulfills established diagnostic criteria for HLH or has progressed to advanced organ failure, the incremental value of a risk-stratification tool is likely to diminish, because management should then be driven primarily by established diagnostic and critical-care frameworks.

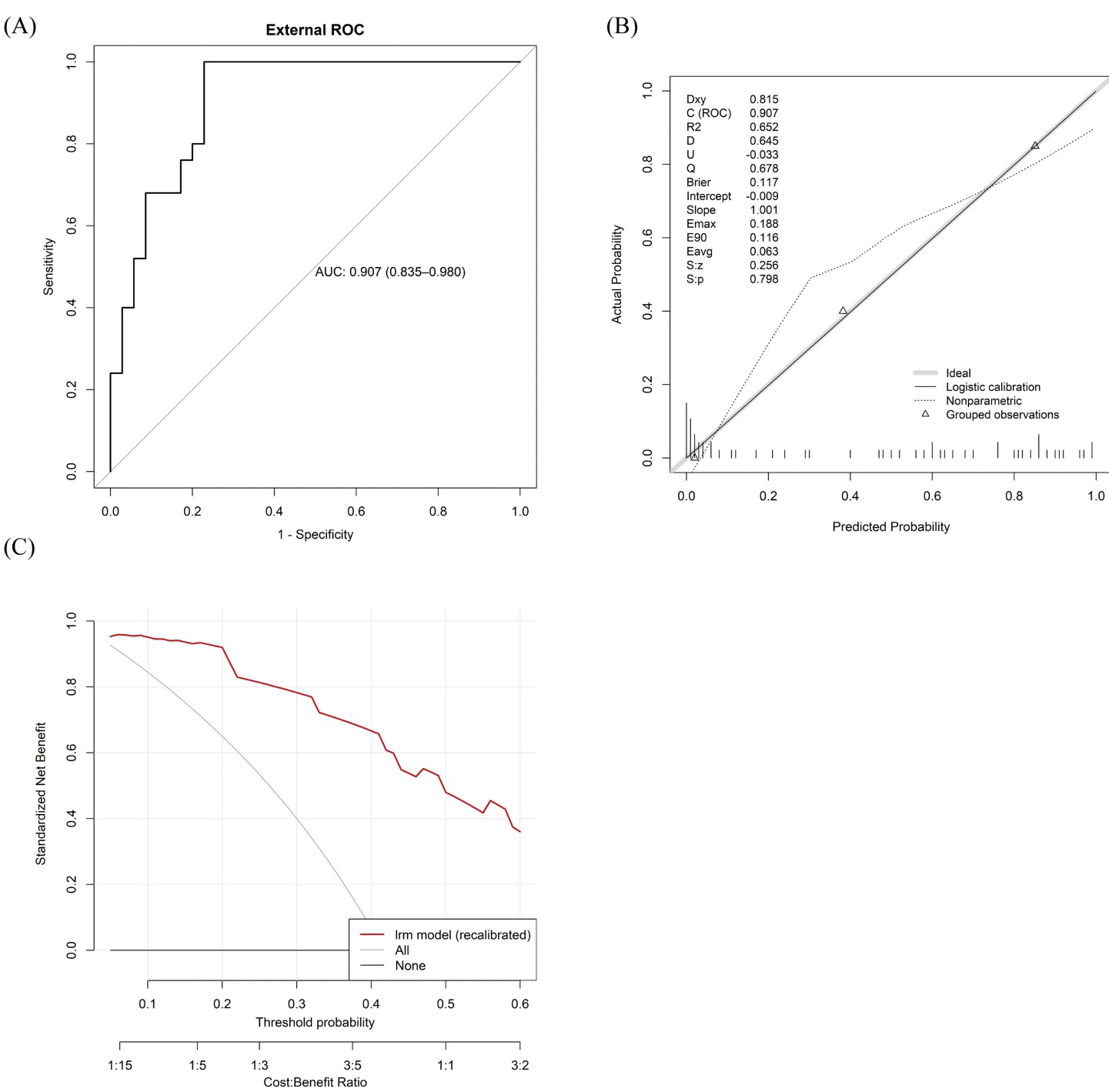

**Fig 4. External validation of the SFTS–HLH prediction model. (A)** Receiver operating characteristic (ROC) curve of the prediction model in the external validation cohort, with the area under the ROC curve (AUC) quantifying discrimination. **(B)** Calibration plot comparing predicted and observed probabilities of secondary HLH in the external cohort. **(C)** Decision-curve analysis in the external validation cohort, showing the net benefit of applying the model across a range of decision thresholds relative to "treat-all" and "treat-none" strategies.

Several limitations merit consideration. First, the retrospective design entails risks of selection bias, information bias, and residual confounding. Second, although data were abstracted from electronic medical records using a standardized protocol, some variables were missing and required imputation (mean or median for continuous measures), which may have attenuated true variability and biased effect estimates. Third, the binary outcome was defined using the entire-course HScore, which incorporates laboratory features that overlap with several candidate predictors. To enhance transparency, we provide detailed outcome accounting for both the early window and the entire clinical course (S1–S3 Tables). Importantly, 83 of 97 patients (85.6%) in the final HLH group already met the HScore threshold at the early evaluation point. Therefore, although the present model may be useful for early risk stratification within a clinically relevant window, the current study does not establish that it identifies HLH earlier than HScore. In addition, several HLH-2004 criteria, including NK-cell activity and soluble CD25, were not assessed in this cohort, and hemophagocytosis was not uniformly evaluated; thus, misclassification of borderline or evolving HLH cannot be entirely excluded. The model was derived at a single provincial center and validated in one external hospital, which may limit generalizability. Moreover, treatment variables and outcomes beyond the in-hospital course were not incorporated, precluding direct assessment of whether early risk identification followed by targeted intervention translates into improved survival.

Future studies should address these limitations. Prospective, multicenter validation in diverse endemic settings is needed to confirm robustness, evaluate transportability, and enable site-specific recalibration where appropriate. Incorporating longitudinal laboratory trajectories and organ support requirements may allow development of a dynamic, time-to-HLH or time-to-deterioration prediction model rather than a single baseline binary risk estimate. To demonstrate clinical utility, it will be essential to link predicted risk to standardized early management pathways—including antiviral therapy, organ protection, and anti-inflammatory or immunoregulatory strategies—and to evaluate patient-centered outcomes such as mortality and organ failure–free survival. Ultimately, implementation of this model in routine triage may improve early risk recognition for sHLH in SFTS; however, its clinical use should be accompanied by ongoing performance surveillance, periodic recalibration, and alignment with evolving standards of care.

## Key points

A six-variable model based on routinely available early clinical and laboratory variables supports risk stratification for evolving hemophagocytic lymphohistiocytosis in severe fever with thrombocytopenia syndrome and may facilitate closer monitoring and earlier diagnostic escalation.

## Supporting information

**S1 Table. HScore distributions within 3 days after virologic confirmation and over the entire clinical course, stratified by final outcome group.** The table reports median (IQR), range, and the proportion meeting the HScore threshold (≥170) for each group.
(DOCX)

**S2 Table. Fulfillment of individual HLH-2004 criteria within 3 days after virologic confirmation, by outcome group.** Results are presented as n/N (%) where N denotes evaluable patients for each criterion; entries recorded as "N" or blank indicate tests not performed and are treated as not evaluable. HLH-2004 criteria are reported for descriptive reference only and were not used for binary outcome classification.
(DOCX)

**S3 Table. Fulfillment of individual HLH-2004 criteria over the entire clinical course, by outcome group.** Results are presented as n/N (%) using evaluable denominators and document non-evaluable tests; HLH-2004 criteria are reported for descriptive reference only and were not used for binary outcome classification.
(DOCX)

**S1 Fig. Flow chart of patient selection for model derivation and validation.** Adult patients with laboratory-confirmed severe fever with thrombocytopenia syndrome (SFTS) admitted during the study period were screened. Patients discharged at their own request within 72 hours of admission or lacking key clinical and laboratory data required within 3 days after virologic confirmation were excluded. The remaining patients were assigned to a derivation cohort, which was randomly split into training and internal validation sets, and an independent external cohort from another center was used for external validation of the prediction model for secondary hemophagocytic lymphohistiocytosis (HLH).
(TIF)

**S2 Fig. Variable selection using least absolute shrinkage and selection operator (LASSO) regression.** (A) Coefficient profiles of candidate predictors as the penalty parameter changes. As the penalty increases, less informative variables shrink toward zero; predictors with non-zero coefficients at the selected penalty were retained for further modelling. (B) Ten-fold cross-validation used to select the optimal penalty parameter ($\lambda$). The dashed line marks the $\lambda$ chosen according to the one-standard-error rule, which favors a simpler model with more stable performance. Retained variables were subsequently cross-checked using the Boruta algorithm before inclusion in the final multivariable logistic model.
(TIF)

**S3 Fig. Comparison of key early clinical and laboratory features between patients with and without HLH.** Selected variables that differed significantly between groups are displayed to illustrate the early phenotypic separation of the HLH and non-HLH cohorts. The figure is intended to complement Table 1 and to provide a visual summary of the stronger inflammatory, coagulation, and organ-injury profile observed in the HLH group.
(TIF)

**S1 Text. TRIPOD checklist.** Transparent Reporting of a multivariable prediction model for Individual Prognosis Or Diagnosis (TRIPOD) checklist for the development and validation of an early risk-stratification model for hemophagocytic lymphohistiocytosis in severe fever with thrombocytopenia syndrome. *BMJ 2024; 385 doi:* https://doi.org/10.1136/bmj-2023-078378 *(Published 16 April 2024).*
(DOCX)

## Author contributions

**Conceptualization:** Yiqiang Hu, Wenting Li, Chuanlong Zhu.

**Data curation:** Yiqiang Hu, Qionghan He, Yuxiao Liu, Jiaqi Kong, Jianting He, Rui Liu.

**Formal analysis:** Yiqiang Hu.

**Funding acquisition:** Chuanlong Zhu.

**Investigation:** Qionghan He, Yuxiao Liu, Jiaqi Kong, Jianting He, Rui Liu.

**Methodology:** Yiqiang Hu, Wenting Li, Chuanlong Zhu.

**Project administration:** Wenting Li, Chuanlong Zhu.

**Resources:** Qionghan He, Yuxiao Liu, Jiaqi Kong, Jianting He, Wenting Li.

**Software:** Yiqiang Hu.

**Supervision:** Wenting Li, Chuanlong Zhu.

**Validation:** Yiqiang Hu.

**Visualization:** Yiqiang Hu.

**Writing – original draft:** Yiqiang Hu.

**Writing – review & editing:** Yiqiang Hu, Qionghan He, Yuxiao Liu, Jiaqi Kong, Jianting He, Rui Liu, Wenting Li, Chuanlong Zhu.

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
