## [Editor Report · Decision Letter 0]

6 Jan 2026

Development and Validation of an Early Risk Prediction Model for HLH in SFTS

Dear Dr. Zhu,

Thank you for submitting your manuscript to PLOS Neglected Tropical Diseases. After careful consideration, we feel that it may have merit but does not fully meet PLOS Neglected Tropical Diseases's publication criteria as it currently stands. Therefore, we invite you to submit a revised version of the manuscript that addresses the points raised during the editorial review process.

Please submit your revised manuscript within by Mar 07 2026 11:59PM. If you will need more time than this to complete your revisions, please reply to this message or contact the journal office at plosntds@plos.org. Please include the following items when submitting your revised manuscript:

* A letter that responds to each point raised by the editor. You should upload this letter as a separate file labeled 'Response to Reviewers'. This file does not need to include responses to any formatting updates and technical items listed in the 'Journal Requirements' section below.

We look forward to receiving your revised manuscript.

Kind regards,

Alan L Rothman, MD

Academic Editor

Stephanie Seifert

Section Editor

Shaden Kamhawi

co-Editor-in-Chief

Paul Brindley

co-Editor-in-Chief

**Additional Editor Comments:**

This manuscript reports an analysis of data from a retrospective cohort of adults with SFTS to develop and validate a multivariable logistic model to discriminate HLH and non-HLH cases based on clinical and laboratory parameters collected within 3 days of virological diagnosis. The authors use appropriate statistical methods to derive and evaluate the model and their results are overall presented in an appropriate fashion. The authors conclude that this model, presented as a nomogram, could support timely patient management through the early identification of HLH cases.

There are several points that would need to be address before this manuscript can be considered for external review:

1. Adjudication of the outcome to be predicted, HLH cases, relied on HScore and HLH-2004 criteria. While these are appropriate tools, they also are heavily weighted on the same laboratory parameters that were used as predictor variables. It is therefore essential that the manuscript provide a full accounting of the outcomes. Specifically, the authors should provide summary statistics of HScore in the two patient groups and report the percentage of subjects in each group that met each of the 8 HLH-2004 criteria and the total number of criteria met.

2. The authors stress the value of their model as an early clinical tool and predictor of HLH. While the model was developed on data collected within 3 days of virological diagnosis, at least half of the subjects reported illness for >7 days prior to admission. The authors should therefore provide information on HScore calculated based on data available at the same early time point and the number of HLH-2004 criteria met at that time point.

In addition to the major points above, the editors offer the following additional initial feedback:

3. Table 1- This table does not clearly state whether the data reflect the entire clinical course of illness or only the first 3 days after virological diagnosis. If the table reflects the entire disease course, a separate table of data during the first 3 days after virological diagnosis (used for the predictive model) would be appropriate. It would be helpful to specify that some parameters, such as myalgia, refer to ever present during illness. For continuous variables, the table should specify whether maximum or minimum values during illness were used.

4. Figure 1- 95% confidence intervals referenced in the legend are not visible.

At this stage, the following Authors/Authors require contributions: Yiqiang Hu, Qionghan He, Yuxiao Liu, Jiaqi Kong, Jianting He, Rui Liu, Wenting Li, and Chuanlong Zhu. Please ensure that the full contributions of each author are acknowledged in the "Add/Edit/Remove Authors" section of our submission form.

2) We have noticed that you have uploaded Supporting Information files, but you have not included a list of legends. Please add a full list of legends for your Supporting Information files after the references list.

**Reviewers' Comments:**

**Figure resubmission:**
---

## [Decision Letter · Decision Letter 1]

1 Apr 2026

Response to Reviewers'. This file does not need to include responses to any formatting updates and technical items listed in the 'Journal Requirements' section below.'. This file does not need to include responses to any formatting updates and technical items listed in the 'Journal Requirements' section below.* A marked-up copy of your manuscript that highlights changes made to the original version. You should upload this as a separate file labeled 'Revised Manuscript with Track Changes'.'.* An unmarked version of your revised paper without tracked changes. You should upload this as a separate file labeled 'Manuscript'.'.If you would like to make changes to your financial disclosure, competing interests statement, or data availability statement, please make these updates within the submission form at the time of resubmission. Guidelines for resubmitting your figure files are available below the reviewer comments at the end of this letter.We look forward to receiving your revised manuscript.Kind regards,Alan L Rothman, MDAcademic EditorPLOS Neglected Tropical DiseasesStephanie SeifertSection EditorPLOS Neglected Tropical Diseases

Shaden Kamhawi

co-Editor-in-Chief

Paul Brindley

co-Editor-in-Chief

**Additional Editor Comments:**The authors provided the detailed information requested in the editorial review of the original submission. As noted by the external reviewers, the significant sample size for a rare and understudied disease is an attractive feature of the manuscript.

Reviewer #3 has recommended specific points that should be addressed in a revised manuscript.

**Reviewers' comments:**Reviewer's Responses to Questions

**Key Review Criteria Required for Acceptance?**

**Methods**

-Are the objectives of the study clearly articulated with a clear testable hypothesis stated?

-Is the study design appropriate to address the stated objectives?

-Is the population clearly described and appropriate for the hypothesis being tested?

-Is the sample size sufficient to ensure adequate power to address the hypothesis being tested?

-Were correct statistical analysis used to support conclusions?

-Are there concerns about ethical or regulatory requirements being met?

Reviewer #1: Please specify the duration of symptoms (median) when data were captured for model development and also initial HLH score calculation.

Reviewer #2: (No Response)

Reviewer #3: Yes

**Results**

-Does the analysis presented match the analysis plan?

-Are the results clearly and completely presented?

-Are the figures (Tables, Images) of sufficient quality for clarity?

Reviewer #1: It appears that authors wanted to develop a model for early prediction. However, by the time initial data were collected for model development, about 85% of the HLH group already had HLH. Hence, it might not be an early prediction.

Reviewer #2: (No Response)

Reviewer #3: Yes

**Conclusions**

-Are the conclusions supported by the data presented?

-Are the limitations of analysis clearly described?

-Do the authors discuss how these data can be helpful to advance our understanding of the topic under study?

-Is public health relevance addressed?

Reviewer #1: (No Response)

Reviewer #2: (No Response)

Reviewer #3: Yes

**Editorial and Data Presentation Modifications?**

Reviewer #1: (No Response)

Reviewer #2: (No Response)

Reviewer #3: (No Response)

**Summary and General Comments**

Reviewer #1: (No Response)

Reviewer #2: (No Response)

Reviewer #3: This paper looks to develop a prognostic indictator of hemophagocytic lymphohistiocytosis in patients infected with SFTS. This study included 249 individuals who had confirmed SFTS and represents a relatively rare group of sufficient size to follow disease progression. The use of Lasso regression and Boruto for feature selection to develop the model is appropriate and the development is thus relatively straightforward. I have but minor comments:

Please spell out acronyms in the title.

Please give some explanation of HScore so that the paper is more broadly accessible to non-clinical readers.

Is there data to report time between symptom onset and positive test return? This presumably would play into treatment algorithms.

A figure showing the differences (or similarities) of the HLH vs. non-HLH patients (corresponding to results lines 136-171) would be of broad interest to readers, I think.

Figure 2 needs a bit more context. Is this for use in clinical setting? For non-clinicians, it is confusing.

Figure S2 needs more explanation for those not familiar with machine learning. Or more labeling, perhaps.

Some discussion on how this would change patient care trajectory would be of interest. At what point is this change in care moot (is there a time window when this needs to be enacted vs. when it would not have an effect)?

How do your results compare to the Korea and Japan (and other China) studies?

PLOS authors have the option to publish the peer review history of their article (what does this mean?). If published, this will include your full peer review and any attached files.). If published, this will include your full peer review and any attached files.). If published, this will include your full peer review and any attached files.). If published, this will include your full peer review and any attached files.

...

Reviewer #1: No

Reviewer #2: **Yes:** Tao ChenTao ChenTao ChenTao Chen

Reviewer #3: No

**Figure resubmission:**While revising your submission, we strongly recommend that you use PLOS’s NAAS tool (https://ngplosjournals.pagemajik.ai/artanalysis) to test your figure files. NAAS can convert your figure files to the TIFF file type and meet basic requirements (such as print size, resolution), or provide you with a report on issues that do not meet our requirements and that NAAS cannot fix.

**Reproducibility:**To enhance the reproducibility of your results, we recommend that authors of applicable studies deposit laboratory protocols in protocols.io, where a protocol can be assigned its own identifier (DOI) such that it can be cited independently in the future. Additionally, PLOS ONE offers an option to publish peer-reviewed clinical study protocols. Read more information on sharing protocols at https://plos.org/protocols?utm_medium=editorial-email&utm_source=authorletters&utm_campaign=protocolsTo enhance the reproducibility of your results, we recommend that authors of applicable studies deposit laboratory protocols in protocols.io, where a protocol can be assigned its own identifier (DOI) such that it can be cited independently in the future. Additionally, PLOS ONE offers an option to publish peer-reviewed clinical study protocols. Read more information on sharing protocols at https://plos.org/protocols?utm_medium=editorial-email&utm_source=authorletters&utm_campaign=protocols

---

## [Editor Report · Decision Letter 2]

12 Apr 2026

Dear Professor Zhu,

We are pleased to inform you that your manuscript 'Development and Validation of an Early Risk-Stratification Model for Hemophagocytic Lymphohistiocytosis in Severe Fever with Thrombocytopenia Syndrome' has been provisionally accepted for publication in PLOS Neglected Tropical Diseases.

Best regards,

Alan L Rothman, MD

Academic Editor

Stephanie Seifert

Section Editor

Shaden Kamhawi

co-Editor-in-Chief

Paul Brindley

co-Editor-in-Chief

---

## [Editor Report · Acceptance letter]

Dear Professor Zhu,

We are delighted to inform you that your manuscript, "Development and Validation of an Early Risk-Stratification Model for Hemophagocytic Lymphohistiocytosis in Severe Fever with Thrombocytopenia Syndrome," has been formally accepted for publication in PLOS Neglected Tropical Diseases.

Best regards,

Shaden Kamhawi

co-Editor-in-Chief

Paul Brindley

co-Editor-in-Chief
